# Cardiovascular diseases worsen the maternal prognosis of COVID-19

Carolina Burgarelli Testa[1], Luciana Graziela de Godoi [2]*, Maria Rita de Figueiredo Lemos Bortolotto [3], Nátaly Adriana Jiménez Monroy[2], Bruna Rodrigues de Mattos[2], Agatha Sacramento Rodrigues[2], Rossana Pulcineli Vieira Francisco[1]

1 Disciplina de Obstetrícia, Departamento de Obstetrícia e Ginecologia, Faculdade de Medicina FMUSP, Universidade de São Paulo, São Paulo, Brazil, 2 DaSLab (Data Science Lab), Department of Statistics, Federal University of Espírito Santo, Vitória/ES, Brazil, 3 Divisão de Clínica Obstétrica, Instituto Central do Hospital das Clínicas HCFMUSP, Faculdade de Medicina FMUSP, Universidade de São Paulo, São Paulo, Brazil

* luciana.godoi@ufes.br

**Data Availability Statement:** Datasets are available in the Kaggle public repository at the URL: https://www.kaggle.com/datasets/agatharodrigues/papermaternal. These data were obtained on May

## Abstract

Cardiovascular diseases (CVD) are a risk factor for severe cases of COVID-19. There are no studies evaluating whether the presence of CVD in pregnant and postpartum women with COVID-19 is associated with a worse prognosis. In an anonymized open database of the Ministry of Health, we selected cases of pregnant and postpartum women who were hospitalized due to COVID-19 infection and with data regarding their CVD status. In the SIVEP GRIPE data dictionary, CVD is defined as "presence of cardiovascular disease", excluding those of neurological and nephrological causes that are pointed out in another field. The patients were divided into two groups according to the presence or absence of CVD (CVD and non-CVD groups). Among the 1,876,953 reported cases, 3,562 confirmed cases of pregnant and postpartum women were included, of which 602 had CVD. Patients with CVD had an older age (p<0,001), a higher incidence of diabetes (p<0,001) and obesity (p<0,001), a higher frequency of systemic (p<0,001) and respiratory symptoms (p<0,001). CVD was a risk factor for ICU admission (p<0,001), ventilatory support (p = 0.004) and oro-tracheal intubation in the third trimester (OR 1.30 CI95%1.04–1.62). The group CVD had a higher mortality (18.9% vs. 13.5%, p<0,001), with a 32% higher risk of death (OR 1.32 CI95%1.16–1.50). Moreover, the risk was increased in the second (OR 1.94 CI95%1.43–2.63) and third (OR 1.29 CI95%1.04–1.60) trimesters, as well as puerperium (OR 1.27 CI95%1.03–1.56). Hospitalized obstetric patients with CVD and COVID-19 are more symptomatic. Their management demand more ICU admission and ventilatory support and the mortality is higher.

## 1. Introduction

Maternal mortality is one of the most sensitive markers of health care because it has multiple economic and social impacts. Brazil has a maternal mortality rate of 59.1/100,000 live births [1], above the values considered acceptable by the World Health Organization, which established as a millennium goal for Brazil rates below 35/100,000 live births [2].

5, 2021 from the following public domain available by the DATASUS: https://opendatasus.saude.gov. br/dataset. All the R code and documentation are available in the GitHub repository at https://github. com/observatorioobstetrico/COVID19_CVD.

**Funding:** This work was supported, in whole or in part, by the Bill & Melinda Gates Foundation [INV-027961]. Under the grant conditions of the Foundation, a Creative Commons Attribution 4.0 Generic License has already been assigned to the Author Accepted Manuscript version that might arise from this submission. This work is also funded by CNPq (Award Number: 445881/2020-8) and FAPES (Award Number: 007/2021). The funders had no role in study design, data collection and analysis, decision to publish, or preparation of the manuscript.

**Competing interests:** The authors have declared that no competing interests exist.

The presence of cardiovascular diseases (CVD) is among the main indirect causes of maternal mortality in the world, mainly in developed countries [3–5]. In Brazil, CVD accounted for 7% to 9.6% of maternal deaths in the years 2011 to 2019 [6].

During the COVID-19 pandemic, maternal mortality has increased significantly in the world. Currently, the postpartum period has been recognized as a risk factor for complications of this infection [7]. By the time this article was written, Brazil had recorded 13,769 cases of Severe Acute Respiratory Syndrome (SARS) and 1,461 deaths (12.3%) due to COVID-19 during pregnancy and the postpartum period [8].

Since the Influenza A (H1N1) pandemic in 2009, the Brazilian Ministry of Health has since established and maintained the Influenza Epidemiological Surveillance Information System (Sistema de Vigilância Epidemiológica de Gripe, SIVEP-Gripe), with the possibility of access to data without the individual identification of the patient (anonymized data), on an open basis [9, 10].

During the COVID-19 pandemic, the notification in the SIVEP-Gripe of cases of patients with SARS hospitalized in all health units in the country has been compulsory. It has enabled the evaluation of diagnosis, demographic and epidemiological data, and the evolution of cases. The case definition of SARS is of an individual with flu-like syndrome with at least two of the following signs and symptoms: fever, chills, sore throat, headache, cough, rhinorrhea, olfactory or taste disorders. They often present with dyspnea/respiratory distress, persistent chest pressure, O2 saturation less than 95% in ambient air, or cyanosis of the lips or face [11].

Studies reported that CVD is an independent risk factor for severe cases of COVID-19 in the general population [12, 13]. However, there are no studies that have evaluated whether the presence of CVD in pregnant and postpartum women with COVID-19 can be considered a risk factor for worse disease evolution, including a higher prevalence of symptoms and a higher risk of ICU admission, orotracheal intubation and death.

## 2. Materials and methods

We conducted a search in the SIVEP-gripe open database, without the individual identification of the patient (anonymized data) on May 5, 2021, on the cases reported in the period between the 8th epidemiological week of 2020 (beginning of the pandemics in Brazil) and the 15th epidemiological week of 2021 (period between February 16, 2020, and April 17, 2021).

Prior approval by institutions board review and patient consent were waived for this study due to an anonymized open-base study, according to Brazilian regulations.

We selected cases of SARS with a confirmed diagnosis by the Ministry of Health of COVID-19 who had been hospitalized. Confirmed cases were those with positivity in laboratory tests (PCR, antigen test or serology), those with a highly suspicious image, and those with a compatible clinic in a patient in contact with a positive case (clinical-epidemiological confirmation) at the time when tests were scarce. After that, we selected pregnant and postpartum women aged 10 to 55 years old with data regarding their CVD status. In the SIVEP GRIPE data dictionary, CVD is defined as "presence of cardiovascular disease", excluding those of neurological and nephrological causes that are pointed out in another field. The patients were divided into two groups according to the presence or absence of CVD (CVD and non-CVD groups).

The evaluated variables were the date of onset of symptoms, age, ethnic group, education, time of pregnancy, risk factors, reported comorbidities (hemopathy, liver disease, asthma, diabetes, neuropathy, pneumopathy, immunodepression, nephropathy and obesity), symptoms, ICU admission, ventilatory support, orotracheal intubation, and outcomes.

The analyses were performed using the statistical software R [14]. The qualitative variables are presented as absolute frequency (n) and percentage (%), while the quantitative variables are presented as mean ± standard deviation (SD).

The comparison between the groups, according to the presence (CVD group) or absence of CVD (non-CVD group) for categorical variables was performed through the X2 test. The correction was conducted by Yates and the Fisher's exact test, when necessary. In addition, the odds ratios (OR) were also calculated with the confidence interval being 95% (95% CI). For continuous variables, the comparison between the independent groups was analyzed by the t-Student test or the Wilcoxon signed-rank test with continuity correction, when appropriate. The results were considered statistically significant when $p < 0.05$.

To control selection biases in non-experimental studies, the CVD and non-CVD groups were balanced with respect to age, ethnic group, obesity, and the presence of diabetes (potential confounding variables in the study). Propensity score matching (PSM) was used for estimating and assessing balancing weights of the observations to make two balanced groups through Inverse Probability of Treatment Weighting Method (IPTW). Logistic regression was the method used to create the propensity score weights and the Average Treatment Effect (ATE) is estimated for treatment effects based on IPTW. Analyses related to the comparison of symptoms and outcomes in the groups with and without CVD were redone considering the PSM weights. PSM was carried out with the R Weightlt package [15, 16]. The groups were balanced in relation to age, ethnic group, obesity and diabetes, as can be seen in the GitHub repository at https://github.com/observatorioobstetrico/COVID19_CVD.

## 3. Results

In the period between the 8th epidemiological week of 2020 and the 15th epidemiological week of 2021 in a consultation held on May 5, 2021, we found 1,876,953 reported cases of COVID-19. We selected only the confirmed and hospitalized cases of COVID-19 in pregnant and postpartum women aged 10 to 55 years who had in their notification the information about the presence or absence of CVD (Fig 1). Among all the cases included, 3392 (95.2%) presented a positive laboratory test, 126 (3.5%) were considered confirmed through imaging, and 44 (1.2%) were considered positive when clinically and epidemiologically confirmed. The final sample size consisted of 3,562 cases, 602 (17%) of which in CVD group and 2,960 (83%) in the non-CVD group.

The absolute frequency and percentage growth rate of COVID-19 cases according to the month of the first symptom in the CVD and non-CVD groups are shown in Fig 2. It is important to note that the growth rate of COVID-19 cases for a specific month of the first symptoms is calculated with respect to the month immediately preceding and the fixed group (CVD or non-CVD).

We noticed that the absolute numbers were increased in both groups from March to May 2020, with a high level until July. Furthermore, there was a fall between August and October and a new increase until March 2021. Note that, if we consider the first wave as the period from March to October 2020, the average growth rate of COVID-19 cases calculated using geometric mean was 11,27% for the CVD group and 23,20% for the non-CVD group. In the second wave, from October 2020 to April 2021, the average growth rate of COVID-19 cases was 36,79% and 29,82% for the CVD and non-CVD groups, respectively. These numbers show us that in the 2nd wave, the virus was highly transmissible in both groups (CVD and non-CVD) in comparison with the first wave and, in particular, more aggressive in the CVD patients since they had a higher rate of contamination.

The characteristics of the studied population are presented in Table 1. Patients with CVD were older (34.76 ± 8.52 vs. 30.00 ± 7.43 years, $p < 0.001$), more often aged over 35 years (49.8% vs. 28.0%, $p < 0.001$). When we evaluated the presence of comorbidities, patients with CVD disease had a higher percentage of associated diseases ($p < 0.001$). The prevalence of

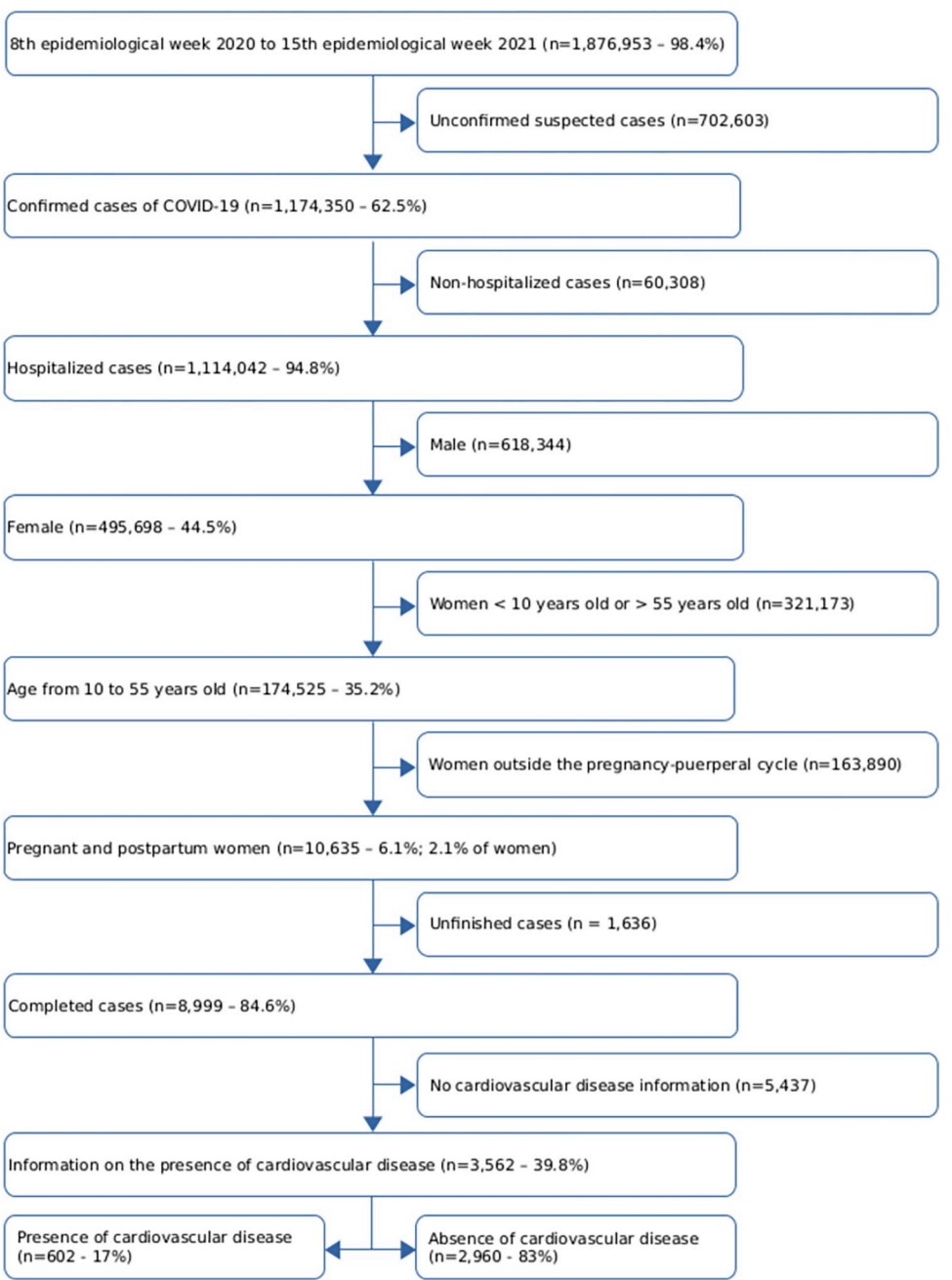

**Fig 1. Case selection flowchart.**

diabetes (33.6% vs. 11.4%, p<0,001), pneumopathy (3.0% vs. 1.1%, p = 0.002), nephropathy (4.4% vs. 1.2%, p<0,001), and obesity (Body Mass Index—BMI>30) (23.4% vs. 8.8%, p<0,001) were higher in CVD group when compared with non-CVD group.

The evaluation of the presence of symptoms (Table 2) showed that patients with CVD were more frequently symptomatic, with the presence of at least one symptom among those evaluated in 96.3% of cases (96.3% vs. 91.3%, p<0,001). Systemic symptoms of fever (61.7% vs. 55.8%, p = 0.011), cough (75.4% vs. 65.6%, p<0.01), anosmia (25.8% vs. 18.6%, p = 0.002) and ageusia (26.0% vs. 17.9%, p<0,001) were more prevalent in this group, as were respiratory symptoms.

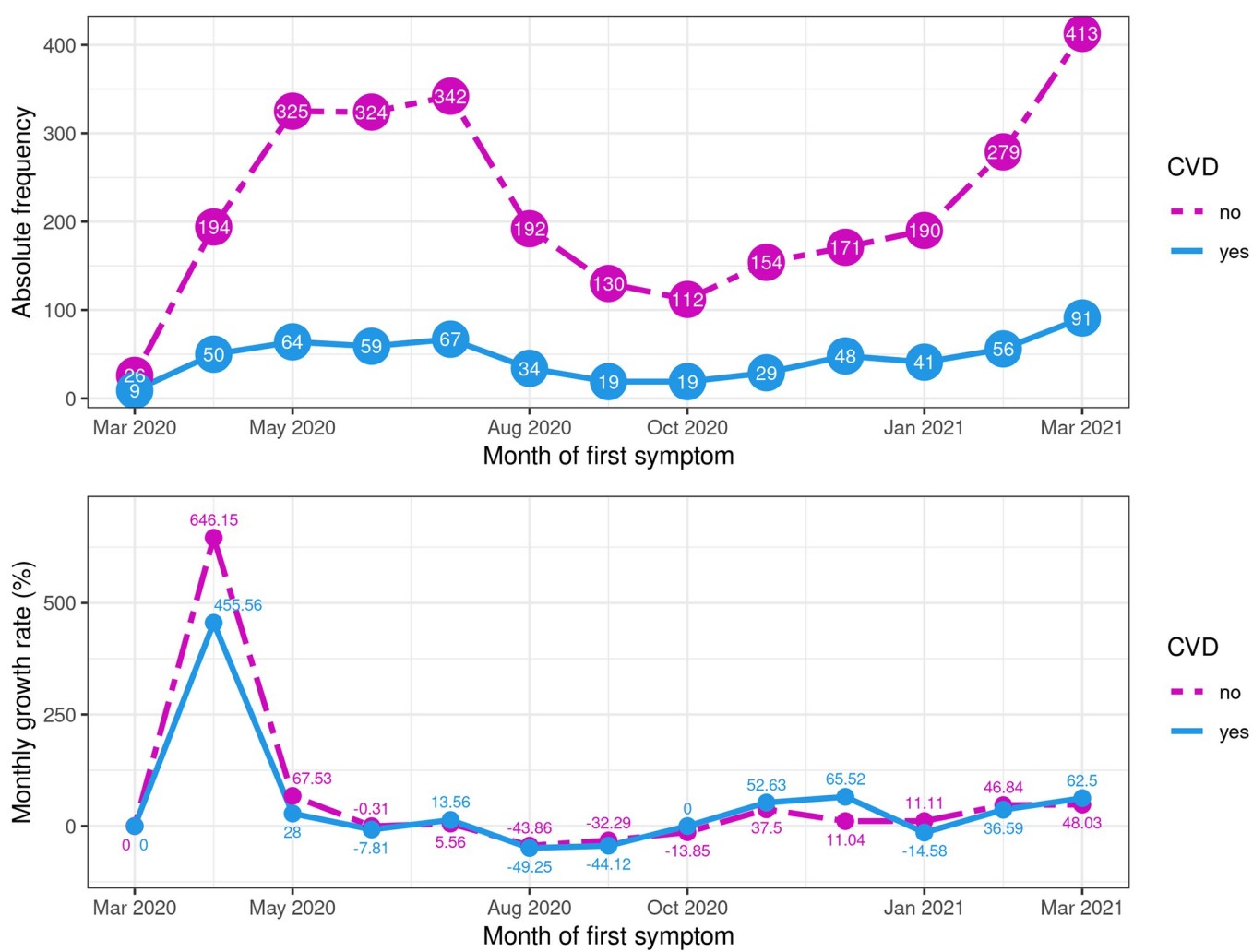

**Fig 2. Absolute frequency and percentage of monthly growth rate of COVID-19 cases according to the month of the first symptom in the CVD and non-CVD groups.**

After applying the Propensity Score Matching (PSM) balancing technique, we found that patients with CVD had an increased risk of dyspnea (OR 1.64 CI95%1.49–1.80), respiratory distress (OR 1.31 CI95%1.19–1.44), and O2 saturation below 95% (OR 1.59 CI95%1.45–1.75). The presence of at least one respiratory symptom affected 83.7% of this group, while there were only 68.1% without CVD (OR 1.68 CI95%1.51–1.88), as well as the presence of a greater number of associated respiratory symptoms that was more frequent in patients with CVD (p<0.001) (Tables 2 and 3).

The presence of CVD was a risk factor for ICU admission (p<0,001). After PSM evaluation, we found a 22% higher risk of admission in the CVD group (OR 1.22 CI95%1.10–1.35), and the risk was increased in the first (OR 1.92 CI95%1.17–3.15) and second (OR 1.34 CI95%1.05–1.71) trimester and in the puerperium period (OR 1.30 CI95%1.08–1.55). Similarly, we found that patients with CVD more often required ventilatory support, both noninvasive and invasive (p = 0.004). There was a greater need for orotracheal intubation of CVD patients in the third trimester of pregnancy (OR 1.30 CI95%1.04–1.62). The length of ICU stay was not statistically different after analysis of the groups via PSM (Table 4).

**Table 1. Demographic and clinical characteristics of pregnant and puerperium women hospitalized with COVID-19 confirmed, according to the presence or absence of cardiovascular disease—Brazil, 02/16/2020 to 04/17/2021.**

| FEATURE | CVD group | Non-CVD group | p |
|---|---|---|---|
| **AGE (YEARS)** | | | |
| Average ± SD | 34.76 ± 8.52 | 30.00 ± 7.43 | <0.001 |
| **AGE GROUP (YEARS)** | **n (%)** | **n (%)** | |
| <20 | 10 (1.7) | 237 (8.0) | <0.001 |
| 20–34 | 292 (48.5) | 1895 (64.0) | |
| ≥35 | 300 (49.8) | 828 (28.0) | |
| **ETHNIC GROUP** | **n (%)** | **n (%)** | |
| Asian | 2 (0.4) | 24 (0.9) | 0.007 |
| White | 214 (42.2) | 975 (37.9) | |
| Indigenous | 1 (0.2) | 30 (1.2) | |
| Mixed | 244 (48.1) | 1375 (53.5) | |
| Black | 46 (9.1) | 167 (6.5) | |
| **EDUCATION** | **n (%)** | **n (%)** | |
| No schooling | 2 (0.7) | 11 (0.7) | 0.241 |
| 1st to 5th grade | 33 (12.2) | 123 (7.9) | |
| 6th to 9th grade | 52 (19.2) | 307 (19.7) | |
| High school | 140 (51.7) | 857 (55.0) | |
| College | 44 (16.2) | 260 (16.7) | |
| **MOMENT OF PREGNANCY** | **n (%)** | **n (%)** | |
| 1st trimester | 34 (5.6) | 161 (5.4) | <0.001 |
| 2nd trimester | 120 (19.9) | 471 (15.9) | |
| 3rd trimester | 248 (41.2) | 1199 (40.5) | |
| Gestational age ignored | 30 (5.0) | 84 (2.8) | |
| Puerperium | 170 (28.2) | 1045 (35.3) | |
| **RISK FACTORS AND COMORBIDITIES** | **n (%)** | **n (%)** | |
| Hemopathy (n = 3387) | 9 / 437 (2.1) | 31 / 2950 (1.1) | 0.113 |
| Hepatopathy (n = 3364) | 5 / 431 (1.2) | 12 / 2933 (0.4) | 0.056 |
| Asthma (n = 3369) | 40 / 436 (9.2) | 199 / 2933 (6.8) | 0.087 |
| Diabetes (n = 3417) | 158 / 470 (33.6) | 337/2947 (11.4) | <0.001 |
| Neuropathy (n = 3374) | 9 / 431 (2.1) | 28 / 2943 (1.0) | 0.062 |
| Pneumopathy (n = 3373) | 13 / 437 (3.0) | 31 / 2936 (1.1) | 0.002 |
| Immunosuppression (n = 3365) | 15/ 433 (3.5) | 58 / 2932 (2.0) | 0.071 |
| Nephropathy (n = 3359) | 19/ 433 (4.4) | 36 / 2926 (1.2) | <0.001 |
| Obesity BMI>30 (n = 3355) | 105 / 449 (23.4) | 255 / 2906 (8.8) | <0.001 |
| **NUMBER OF COMORBIDITIES (n = 3460)** | **n (%)** | **n (%)** | |
| 0 | 234 (46.8) | 2126 (71.8) | <0.001 |
| 1 | 186 (37.2) | 701 (23.7) | |
| 2 | 64 (12.8) | 116 (3.9) | |
| ≥3 | 16 (3.2) | 17 (0.6) | |

Among the patients evaluated, 3,047 had a positive outcome (cure), while 515 died (14.4%). Mortality was higher in the CVD group (18.9% vs. 13.5%, p<0,001). In Fig 3, considering the group (CVD or non-CVD) and the month of first symptom fixed, the notation *a/ b* indicates that of the *b* individuals with the first symptoms in a specific month, *a* died. As shown in Fig 3, the percentage of mortality among patients with CVD was higher in virtually all months evaluated, with an increasing trend in the proportion of deaths since August 2020 and stabilization

**Table 2. Symptoms of COVID-19 in pregnant and puerperal women with cardiovascular disease, hospitalized with COVID-19 confirmed, according to the presence or absence of cardiovascular disease—Brazil, 02/16/2020 to 04/17/2021.**

| SYMPTOMS | CVD group n (%) | Non-CVD group n (%) | P | OR (95% CI) | WEIGHTED PROPENSITY SCORE MATCHING | |
|---|---|---|---|---|---|---|
| | | | | | P | OR (95% CI) |
| At least 1 symptom (n = 3537) | 574 / 596 (96.3) | 2685 / 2941 (91.3) | <0.001 | 2.49 (1.59–3.88) | <0.001 | 2.28 (1.85–2.81) |
| **SYSTEMIC SYMPTOMS** | | | | | | |
| Fever (n = 3460) | 337 / 546 (61.7) | 1625 / 2914 (55.8) | 0.011 | 1.28 (1.06–1.54) | 0.228 | 1.06 (0.97–1.17) |
| Cough (n = 3477) | 424 / 562 (75.4) | 1913 / 2915 (65.6) | <0.001 | 1.61 (1.31–1.98) | 0.138 | 1.08 (0.98–1.19) |
| Sore Throat (n = 3352) | 124 / 491 (25.3) | 616 / 2861 (21.5) | 0.075 | 1.23 (0.99–1.54) | 0.117 | 1.10 (0.98–1.23) |
| Anosmia (n = 2404) | 87 / 337 (25.8) | 384 / 2067 (18.6) | 0.002 | 1.53 (1.17–1.99) | 0.002 | 1.25 (1.08–1.43) |
| Ageusia (n = 2401) | 88/ 338 (26.0) | 369 / 2063 (17.9) | <0.001 | 1.62 (1.24–2.11) | <0.001 | 1.27 (1.11–1.47) |
| **RESPIRATORY SYMPTOMS** | | | | | | |
| Dyspnea (n = 3469) | 401 / 555 (72.3) | 1553 / 2914 (53.3) | <0.001 | 2.28 (1.87–2.79) | <0.001 | 1.64 (1.49–1.80) |
| Respiratory distress (n = 3431) | 316 / 529 (59.7) | 1361 / 2902 (46.9) | <0.001 | 1.68 (1.39–2.03) | <0.001 | 1.31 (1.19–1.44) |
| $O_2$ saturation <95% (n = 3408) | 301 / 532 (56.6) | 1003 / 2876 (34.9) | <0.001 | 2.43 (2.02–2.94) | <0.001 | 1.59 (1.45–1.75) |
| Fatigue (n = 2411) | 104 /342 (30.4) | 460 / 2069 (22.2) | 0.001 | 1.53 (1.19–1.97) | 0.171 | 1.10 (0.96–1.25) |
| One or more respiratory symptoms (n = 3505) | 484 / 578 (83.7) | 1992 / 2927 (68.1) | <0.001 | 2.42 (1.91–3.05) | <0.001 | 1.68 (1.51–1.88) |
| **GASTROINTESTINAL SYMPTOMS** | | | | | | |
| Diarrhea (n = 3365) | 63 / 484 (13) | 324 / 2881 (11.2) | 0.293 | 1.18 (0.88–1.58) | 0.616 | 0.96 (0.82–1.12) |
| Emesis (n = 3352) | 58 / 481 (12.1) | 282 / 2871 (9.8) | 0.155 | 1.26 (0.93–1.70) | 0.308 | 1.09 (0.93–1.27) |
| Abdominal pain (n = 2387) | 36 / 327 (11.0) | 183 / 2060 (8.9) | 0.250 | 1.27 (0.87–1.85) | 0.336 | 0.90 (0.73–1.10) |

at higher levels in March and April 2021. After balancing by PSM, the group with CVD presented a 32% higher risk of death (OR 1.32 CI95%1.16–1.50). The distribution by trimesters of pregnancy and puerperium showed a similar risk of death in the first trimester and increased in the second (OR 1.94 CI95%1.43–2.63) and third (OR 1.29 CI95%1.04–1.60) trimesters. The same was found in the puerperium group (OR 1.27 CI95%1.03–1.56) in patients in the CVD group.

## 4. Discussion

Patients with CVD presented a higher risk of admission to the ICU, need for ventilatory support in all phases of the pregnancy-puerperal cycle studied and orotracheal intubation in the third trimester. In addition, they had a higher mortality rate in the period analyzed, with a 32% higher risk of death than patients without CVD. The evaluation of moment of pregnancy and puerperium cycle showed an increased risk of death in the second and third trimesters and in the puerperium.

**Table 3. Respiratory symptoms of COVID-19 in pregnant and postpartum women with cardiovascular disease, hospitalized with confirmed COVID-19, according to the presence or absense of cardiovascular disease–Brazil, 02/16/2020 to 04/17/2021.**

| RESPIRATORY SYMPTOMS | CVD group n (%) | Non-CVD group n (%) | p | WEIGHTED PROPENSITY SCORE MATCHING |
|---|---|---|---|---|
| | | | | p |
| No symptom | 94 (16.3) | 935 (31.9) | <0.001 | <0.001 |
| 1 symptom | 116 (20.1) | 640 (21.9) | | |
| 2 symptoms | 151 (26.1) | 528 (18.0) | | |
| 3 symptoms | 164 (27.4) | 615 (21.0) | | |
| 4 symptoms | 53 (9.2) | 209 (7.1) | | |

**Table 4. Comparison of outcomes in pregnant and postpartum women with cardiovascular disease, hospitalized with confirmed COVID-19, according to the presence or absence of cardiovascular disease—Brazil, 02/16/2020 to 04/17/2021.**

| OUTCOME | CVD group | Non-CVD group | p | OR (95%CI) | WEIGHTED PROPENSITY SCORE MATCHING | |
|---|---|---|---|---|---|---|
| | | | | | P | OR (95%CI) |
| **ICU ADMISSION** | **n (%)** | **n (%)** | | | | |
| Total (n = 3410) | 209 / 565 (37.0) | 802 / 2845 (28.2) | <0.001 | 1.50 (1.24–1.81) | <0.001 | 1.22 (1.10–1.35) |
| **ICU ADMISSION BY GESTATIONAL OR POST-PARTUM MOMENT** | **n (%)** | **n (%)** | | | | |
| 1st trimester (n = 188) | 13 / 31 (41.9) | 36 / 157 (22.9) | 0.048 | 2.43 (1.09–5.43) | 0.014 | 1.92 (1.17–3.15) |
| 2nd trimester (n = 571) | 47 / 113 (41.6) | 143 / 458 (31.2) | 0.047 | 1.57 (1.03–2.39) | 0.021 | 1.34 (1.05–1.71) |
| 3rd trimester (n = 1386) | 74 / 233 (31.8) | 270 / 1153 (23.4) | 0.009 | 1.52 (1.12–2.07) | 0.091 | 1.16 (0.98–1.37) |
| Puerperium (n = 1174) | 67 / 161 (41.6) | 335 / 1013 (33.1) | 0.042 | 1.44 (1.03–2.03) | 0.005 | 1.30 (1.08–1.55) |
| **DAYS OF ICU STAY** | **Average ± SD** | **Average ± SD** | | | | |
| Average ± SD | 13.46 ± 14.49 | 10.53 ± 10.73 | 0.029 | | 0.198 | |
| **VENTILATORY SUPPORT** | **n (%)** | **n (%)** | | | | |
| No (n = 1696) | 226 / 555 (40.7) | 1470 / 2790 (52.7) | <0.001 | | 0.004 | |
| Non-Invasive (n = 1096) | 219 / 555 (39.5) | 877 / 2790 (31.4) | | | | |
| Invasive (n = 553) | 110 / 555 (19.8) | 443 / 2790 (15.9) | | | | |
| **OROTRACHEAL INTUBATION FOR GESTATIONAL OR PUERPERIUM MOMENT** | **n (%)** | **n (%)** | | | | |
| 1st trimester (n = 181) | 6/ 29 (20.7) | 17 / 152 (11.2) | 0.218 | 2.07 (0.74–5.80) | 0.078 | 1.89 (0.98–3.62) |
| 2nd trimester (n = 550) | 23 / 111 (20.7) | 66 / 439 (15.0) | 0.191 | 1.48 (0.87–2.51) | 0.252 | 1.22 (0.89–1.67) |
| 3rd trimester (n = 1363) | 36 / 231 (15.6) | 125 / 1132 (11.0) | 0.066 | 1.49 (1.00–2.22) | 0.023 | 1.30 (1.04–1.62) |
| Puerperium (n = 1159) | 39 / 157 (24.8) | 224 / 1002 (22.4) | 0.556 | 1.15 (0.78–1.70) | 0.609 | 1.06 (0.86–1.30) |
| Total (n = 3345) | 110 / 555 (19.8) | 443 / 2790 (15.9) | 0.026 | 1.31 (1.04–1.65) | 0.165 | 1.10 (0.97–1.25) |
| **OUTCOME—CURE x DEATH** | **n (%)** | **n (%)** | | | | |
| Cure (n = 3047) | 488 / 602 (81.1) | 2559 / 2960 (86.5) | <0.001 | 1.49 (1.19–1.88) | <0.001 | 1.32 (1.16–1.50) |
| Death (n = 515) | 114 / 602 (18.9) | 401 / 2960 (13.5) | | | | |
| **DEATH BY GESTATIONAL OR POSTPARTUM MOMENT** | **n (%)** | **n (%)** | | | | |
| 1st trimester (n = 195) | 4 / 34 (11.8) | 15 / 161 (9.3) | 0.749 | 1.30 (0.40–4.18) | 0.627 | 1.28 (0.62–2.60) |
| 2nd trimester (n = 591) | 30 / 120 (25.0) | 61 / 471 (13.0) | 0.002 | 2.24 (1.37–3.67) | <0.001 | 1.94 (1.43–2.63) |
| 3rd trimester (n = 1447) | 36 / 248 (14.5) | 128 / 1199 (10.7) | 0.104 | 1.42 (0.95–2.12) | 0.024 | 1.29 (1.04–1.60) |
| Puerperium (n = 1215) | 37 / 170 (21.8) | 189 / 1045 (18.1) | 0.300 | 1.26 (0.85–1.87) | 0.031 | 1.27 (1.03–1.56) |
| **INTERVAL BETWEEN SYM-PTOM ONSET AND OUTCOME (DAYS)** | **Average ± SD** | **Average ± SD** | | | | |
| Total | 17.10 ± 12.80 | 16.00 ± 15.8 | <0.001 | | 0.912 | |
| Death | 20.04 ± 14.11 | 20.34 ± 13.86 | 0.633 | | 0.209 | |
| Cure | 16.39 ± 12.37 | 15.30 ± 15.99 | <0.001 | | 0.723 | |

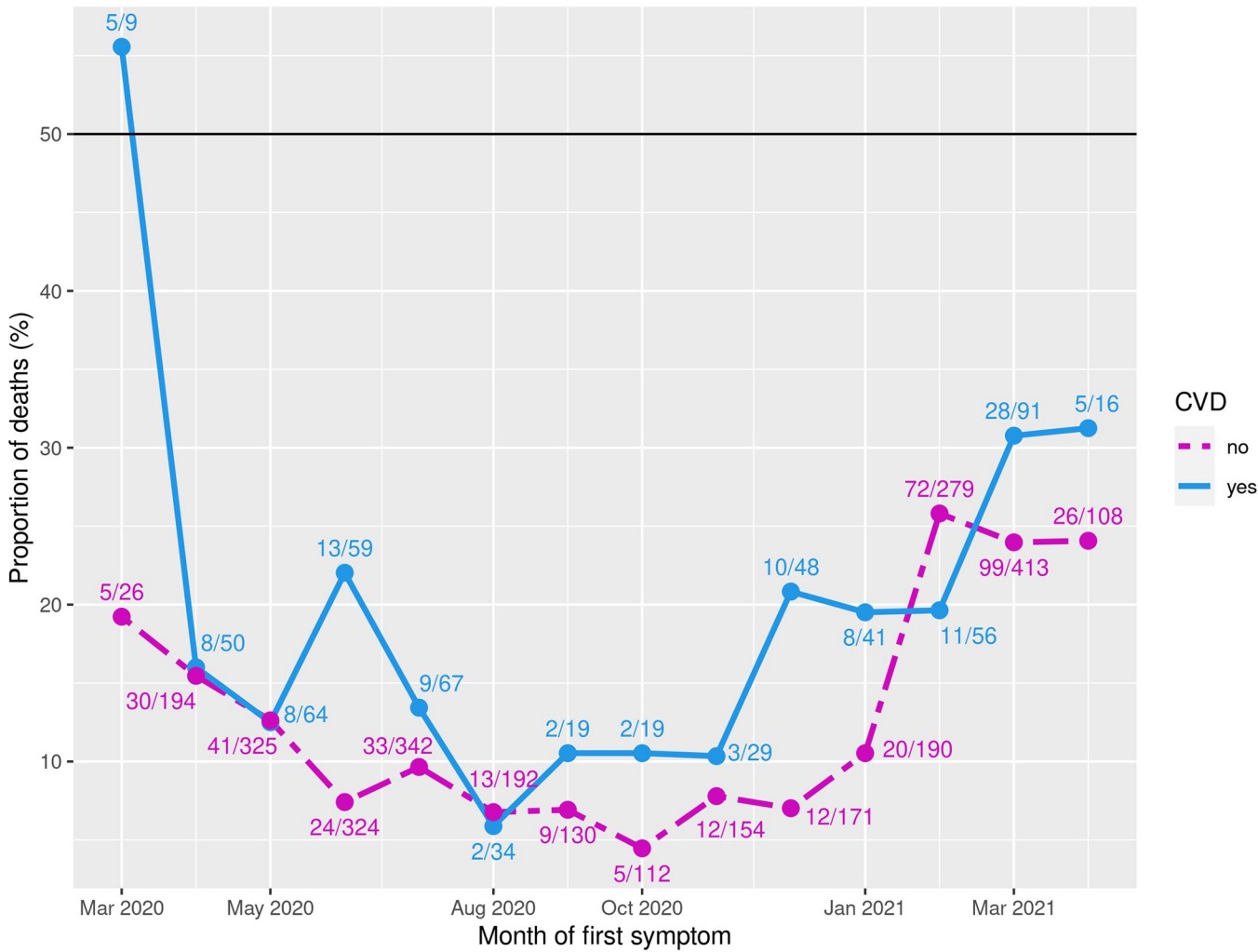

**Fig 3. Percentage of mortality among patients with CVD, fixed the month of first symptom.**

In our population data base, among 3,562 cases of pregnant and puerperal women hospitalized with confirmed COVID-19, we found 602 patients with CVD (17%), a higher incidence than that found in population studies with estimates of around 0.2% to 4% of the pregnant women as carriers of CVD [5, 17]. It is noted that the frequency of CVD in our study was similar to that referred to us and in the North American population study with 8,207 pregnant women with a positive laboratory test for COVID-19, in which 14% of the pregnant women had CVD [18].

The assessment of declared comorbidities showed that patients with CVD had a greater number of associated comorbidities compared to patients without cardiovascular involvement, with a significantly higher incidence of diabetes and obesity, which were used in the Propensity Score Matching (PSM) balancing technique, as well as age and race, to avoid that these factors could impact the results and the comparison between the groups.

Pregnant women with CVD were more symptomatic compared with the control group. They also had more rates of systemic symptoms, such as fever and cough, and respiratory symptoms, such as dyspnea, respiratory distress, and O2 saturation less than 95%. In a systematic review involving 185 obstetric patients evaluated, Narang et al. found 53.2% of patients

presented fever, 33.13% cough, and 13% dyspnea [19]. In our study, the incidence of symptoms was significantly higher, probably because our cohort included only hospitalized patients. In the CVD group, the prevalence of at least one respiratory symptom was 83.7%, with 72.3% of patients presenting with dyspnea, and 56.6% of patients presenting with a saturation drop below 95%. The presence of significantly higher rates of respiratory symptoms in the group of patients with CVD emphasized the importance of adequate clinical diagnosis of complicated COVID-19 infection. Considering that among the main signs and symptoms of cardiac decompensation during pregnancy are cough and respiratory symptoms such as dyspnea, respiratory distress and saturation loss, symptoms among clinical conditions may overlap. In addition, among the main causes of clinical decompensation of a pregnant woman with a pathological cardiac condition is infection [20–23].

ICU admission occurred in 29.6% of the cases in our cohort. This is a rate higher than that found by Narang et al. in a meta-analysis (3.24%) [19] and by Zambrano et al. in a population study with patients who tested positive for COVID-19 (10.5%) [24]. It was observed that when only hospitalized patients were evaluated, the percentage of ICU patients was higher. This was supported by a study by Maeda et al., which involved obstetric patients admitted to a reference hospital for the care of pregnant women with COVID-19 (22.9%) [25]. In patients with CVD, admission to the ICU was found in 37% of the cases evaluated. ICU admission indirectly reflected the case with more severe evolution and was directly related to the need for ventilatory support. In our study, patients with CVD required ventilatory support, both noninvasive and invasive, with an orotracheal intubation rate of 19.8%.

In the face of changes imposed by pregnancy, the cardiovascular system undergoes several adaptations that directly impact the management of severe patients and contribute to the higher mortality of these patients [22, 26]. Maternal mortality in patients with CVD is high, with significant variations according to their baseline CVD, which ranges from 1 to 5%. However, this population always have a higher mortality compared with the general population [4, 26–28]. In the context of the COVID-19 pandemic, the presence of CVD has been recognized as a risk factor for death in several studies and meta-analyses that considered the general population, with incidence of CVD in 37 to 69% of hospitalized cases and lethality rate of 10.5 to 15.5% [29, 30]. However, there is no study in the literature evaluating mortality in obstetric population with CVD in the presence of COVID-19 infection. In our cohort, mortality among pregnant and puerperal women hospitalized with COVID-19 was 14.4%. In the group of patients with CVD, 18.9% of the patients died. The mortality found was higher than those reported in the literature for the general population, which reinforces our hypothesis that pregnant women have a more severe presentation.

Among the greatest difficulties in assisting patients with CVD, COVID-19 infection is the most common infection that presents with respiratory symptoms, which may be secondary to infection or triggered cardiac decompensation. The high proportion of deaths in this group, sustained throughout the months of the pandemic, demonstrate the difficulty of clinical management especially in those with comorbidities.

In addition, we should consider that the patient with CVD is, in general, a chronic patient. During the pandemic period, in which human and material resources were allocated for the treatment of COVID-19, many patients lost their follow-ups and had treatment delays, which might worsen their primary diseases [31].

Mortality in this group, therefore, results from a potentially severe infection in patients at increased risk due to puerperal pregnancy status and CVD, associated with the risks of decompensation of the primary disease, treatment failure, and difficulty in accessing health.

To date, only one series of five cases has been published evaluating the association between the presence of CVD and pregnancy in the presence of COVID-19 infection [32]. We were

worth a population database, which allows us to study a large number of hospitalized cases and with the outcome finalized (3,562 cases evaluated and 609 patients with CVD). So far, this is the largest number of obstetric patients with CVD studied in the face of COVID-19 infection. Among the limitations of the study, we should consider that the completion of the notification system included information about the presence or absence of CVD in 39.58% of the notifications, with a high percentage of loss. The difficulty in population-based use was also portrayed by other researchers. In the USA, for instance, only 5.8% (7162) of 122653 cases reported to the CDC had information about health conditions [33]. In addition, we found limitation on the type of CVD, since the notification form does not allow etiological distinction and interpretation of access to ICU admission in Brazil. This is because only 15% of maternity hospitals in our country have adult ICU beds [34]. Another weakness of our study is the lack of information on the evolution of CVD before COVID-19 infection, including treatments used, duration of illness and quality of follow-up.

## 5. Conclusions

Obstetric patients with CVD and COVID-19 who are hospitalized are more symptomatic. They require ICU hospitalization and ventilatory support. Moreover, they have higher mortality rate than COVID-19 patients without CVD. Mortality in this group may be a consequence of the association between the severity of COVID-19 infection in pregnant women associated with the presence of CVD (which are, by themselves, are recognized risk factor for ICU admission, orotracheal intubation, and death during pregnancy and puerperium). Further studies are needed in this group of patients to determine the main associated risk factors, analyze individually the etiology of CVD, and thus allow improvements in care and consequent reduction in maternal mortality rates.

## Author Contributions

**Conceptualization:** Carolina Burgarelli Testa, Luciana Graziela de Godoi, Maria Rita de Figueiredo Lemos Bortolotto, Agatha Sacramento Rodrigues, Rossana Pulcineli Vieira Francisco.

**Data curation:** Carolina Burgarelli Testa, Luciana Graziela de Godoi, Nátaly Adriana Jiménez Monroy, Bruna Rodrigues de Mattos, Agatha Sacramento Rodrigues, Rossana Pulcineli Vieira Francisco.

**Formal analysis:** Carolina Burgarelli Testa, Luciana Graziela de Godoi, Nátaly Adriana Jiménez Monroy, Bruna Rodrigues de Mattos, Agatha Sacramento Rodrigues, Rossana Pulcineli Vieira Francisco.

**Funding acquisition:** Agatha Sacramento Rodrigues, Rossana Pulcineli Vieira Francisco.

**Investigation:** Carolina Burgarelli Testa, Luciana Graziela de Godoi, Agatha Sacramento Rodrigues, Rossana Pulcineli Vieira Francisco.

**Methodology:** Carolina Burgarelli Testa, Luciana Graziela de Godoi, Maria Rita de Figueiredo Lemos Bortolotto, Nátaly Adriana Jiménez Monroy, Agatha Sacramento Rodrigues, Rossana Pulcineli Vieira Francisco.

**Supervision:** Agatha Sacramento Rodrigues, Rossana Pulcineli Vieira Francisco.

**Writing – original draft:** Carolina Burgarelli Testa.

**Writing – review & editing:** Carolina Burgarelli Testa, Luciana Graziela de Godoi, Maria Rita de Figueiredo Lemos Bortolotto, Nátaly Adriana Jiménez Monroy, Agatha Sacramento Rodrigues, Rossana Pulcineli Vieira Francisco.

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
