## [Decision Letter · Decision Letter 0]

5 May 2022

PONE-D-22-08926Cardiovascular diseases worsen the maternal prognosis of COVID-19PLOS ONE

Dear Dr. Luciana Graziela de Godoi,

Thank you for submitting your manuscript to PLOS ONE. After careful consideration, we feel that it has merit but does not fully meet PLOS ONE’s publication criteria as it currently stands. Therefore, we invite you to submit a revised version of the manuscript that addresses the points raised during the review process.

Please carefully consider all of the reviewers’ comments, criticisms and suggestions below particularly confirmed diagnosis of COVID-19 and the definition of cardiovascular diseases and several others that are required to define clearly. In addition, please carefully follow all comments concerning methodology, statistical analysis, results, conclusions and limitations of the study.

We look forward to receiving your revised manuscript.

Kind regards,

Vipa Thanachartwet, M.D.

Academic Editor

PLOS ONE

Journal Requirements:

"This research was funded by Bill & Melinda Gates Foundation, National Council for Scientific and Technological Development (CNPQ - Brasília, Brazil), and Foundation for Research and Innovation Support of the State of Espírito Santo (FAPES - Espírito Santo, Brazil) are funding source."

We note that you have provided funding information. However, funding information should not appear in the Funding section or other areas of your manuscript. We will only publish funding information present in the Funding Statement section of the online submission form. 

"This work was supported, in whole or in part, by the Bill & Melinda Gates Foundation (INV-027961). Under the grant conditions of the Foundation, a Creative Commons Attribution 4.0 Generic License has already been assigned to the Author Accepted Manuscript version that might arise from this submission. This work is also funded by CNPq (Award Number: 445881/2020-8) and FAPES (Award Number: 007/2021)."

Reviewers' comments:

Reviewer's Responses to Questions

**Comments to the Author**

1. Is the manuscript technically sound, and do the data support the conclusions?

Reviewer #1: Partly

Reviewer #2: Yes

Reviewer #3: Partly

2. Has the statistical analysis been performed appropriately and rigorously? 

Reviewer #1: Yes

Reviewer #2: Yes

Reviewer #3: No

3. Have the authors made all data underlying the findings in their manuscript fully available?

Reviewer #1: Yes

Reviewer #2: Yes

Reviewer #3: Yes

4. Is the manuscript presented in an intelligible fashion and written in standard English?

Reviewer #1: Yes

Reviewer #2: Yes

Reviewer #3: Yes

5. Review Comments to the Author

Reviewer #1: This is an observational study with a large sample size on the evaluation whether the presence of CVD in pregnant women and in postpartum women with COVID-19 is associated with a worse prognosis. Authors used an anonymized open database. They concluded that “Hospitalized obstetric patients with CVD and COVID-19 are more symptomatic.” and that “Their management demand more ICU admission and ventilatory support and the mortality is higher.”

I think that this is a publishable work, but I have some points that need reconsideration.

In the abstract, as in the whole text, there must be clearer distinctions for the groups studied and compared.

In the introduction section, the rationale should be revisited. The aim of the study should be clearly stated in the last sentence.

In the materials and methods, parts from there should be moved in the introduction section; authors should make clear what their comparisons should be and the exact methods; this should be followed in all the next sections, for the reader to understand exactly the correlations. The hospitalized COVID positive patients could be divided according to trimesters (as the results show afterwards), to their state (for e.g. incubated or not e.t.c), days of hospitalization, of NICU stay etc.; other comorbidities should be discriminated – for example a subgroup analysis there might be of essence.

The results should contain the exact results of the comparisons.

The limitations of the study should be presented.

The conclusions should adhere to the exact aim, findings and type of the study presented.

Reviewer #2: The authors present convincing data on the worse prognosis of pregnant women with COVID-19 infection and cardiovascular diseases (CVD). However, some points need to be clarified:

1. Could the authors present both groups of women with the WHO classification of symptoms (asymptomatic, mild-moderate, and severe infection).

2. Lack of information on the baseline and follow-up of CVD patients. Please provide how many years CVD before COVID-19 infection, treatment for CVD (yes or no) and type of treatment, lost of follow-up during COVID-19 infection, treatment delay during COVID-19 infection.

3. Please explain why the propensity score matching was used as the method for balancing weights.

Reviewer #3: The paper has major problems, denying it the right to be published. The main problems are (1) The large proportion of missing data of the patients’ background diseases, which can highly influence the results. (2) The lack of information on CVD definition. (3) The diagnosis method of COVID-19, which is clinically based and not confirmed by PCR in my understanding.

- English editing is required.

- Abstract: “cardiovascular diseases” definition required. Which diseases were included?

- Methods:

o “The case definition of SARS is of an individual with flu-like syndrome”- A confirming PCR test was not obligatory for diagnosis?

o Children were included (age 10-18), in my opinion this population deserves a sub-analysis.

o “To control selection biases in non-experimental studies, the CVD and non-CVD groups were balanced with respect to..”- “balanced” needs to be written “adjusted”.

- Results:

o “We selected only the confirmed and hospitalized cases of COVID-19”- The confirmation methods needs to be specified in details in the “Methods” section.

o 602 of 3,562 cases had CVD? It is a young population, so it seems to me to be over-estimated. Please, again, specify the definition of CVD in this study. Which ICD codes were included?

o “We noticed that the absolute numbers were increased in both groups from March to May 2020, with a high level until July.” – What absolute numbers?

o “This was different from what happened in the 2nd wave in which CVD patients were mostly affected.” – How did you conclude this? Did you calculate the proportion out of the entire “maternal” population (including non-COVID)?

o In table 1-

What is included under “Hemopathy”? “Hepatopathy”? “Pneumopathy”? “Immunosuppression”? “Nephropathy”?

How did you define obesity?

o What was the absolute mortality rate in every group?

- Conclusions- Most of the conclusions cannot be referred to without the above information. That is why, in fact, the paper is rejected.

- Limitation-

o “limitations of the study, we should consider that the completion of the notification system included information about the presence or absence of CVD in 39.58% of the notifications, with a high percentage of loss.” Can you clarify this sentence? Is approximately 60% of the patients’ background status missing?

6. PLOS authors have the option to publish the peer review history of their article (what does this mean?). If published, this will include your full peer review and any attached files.

Reviewer #1: **Yes: **Charalampos S Siristatidis

Reviewer #2: No

Reviewer #3: No

---

## [Author Response · Author response to Decision Letter 0]

18 Jun 2022

We are grateful for the reviewer’s careful reading of the paper, as well as the editor, which has produced many helpful comments and greatly improved our presentation. Responses to each point raised by the academic editor and reviewers are attached as a separate file labeled 'Response to Reviewers'. Thank you in advance for considering this revision. 

Best regards,

Luciana Graziela de Godoi

---

## [Decision Letter · Decision Letter 1]

19 Jul 2022

Cardiovascular diseases worsen the maternal prognosis of COVID-19

PONE-D-22-08926R1

Dear Dr. Luciana Graziela de Godoi,

We’re pleased to inform you that your manuscript has been judged scientifically suitable for publication and will be formally accepted for publication once it meets all outstanding technical requirements.

Kind regards,

Vipa Thanachartwet, M.D.

Academic Editor

PLOS ONE

Additional Editor Comments (optional):

All issues were revised according to the reviewers' comments and suggestions.

Reviewers' comments:

Reviewer's Responses to Questions

**Comments to the Author**

1. If the authors have adequately addressed your comments raised in a previous round of review and you feel that this manuscript is now acceptable for publication, you may indicate that here to bypass the “Comments to the Author” section, enter your conflict of interest statement in the “Confidential to Editor” section, and submit your "Accept" recommendation.

Reviewer #1: (No Response)

2. Is the manuscript technically sound, and do the data support the conclusions?

Reviewer #1: (No Response)

3. Has the statistical analysis been performed appropriately and rigorously? 

Reviewer #1: (No Response)

4. Have the authors made all data underlying the findings in their manuscript fully available?

Reviewer #1: (No Response)

5. Is the manuscript presented in an intelligible fashion and written in standard English?

Reviewer #1: (No Response)

6. Review Comments to the Author

Reviewer #1: (No Response)

7. PLOS authors have the option to publish the peer review history of their article (what does this mean?). If published, this will include your full peer review and any attached files.

Reviewer #1: No

---

## [Editor Report · Acceptance letter]

2 Aug 2022

PONE-D-22-08926R1 

Cardiovascular diseases worsen the maternal prognosis of COVID-19 

Dear Dr. Godoi:

I'm pleased to inform you that your manuscript has been deemed suitable for publication in PLOS ONE. Congratulations! Your manuscript is now with our production department. 

Kind regards, 

on behalf of

Associate Professor Vipa Thanachartwet 

Academic Editor

PLOS ONE